# Tackling Paradoxes and Double Binds for a Healthier Workplace: Insights from the Early COVID-19 Responses in Quebec and Ontario

Daniel Côté [1,2,*], Amelia León [1], Ai-Thuy Huynh [1], Jessica Dubé [1,3], Ellen MacEachen [4], Pamela Hopwood [4], Marie Laberge [5], Samantha Meyer [4], Shannon Majowicz [4], Meghan K. Crouch [4] and Joyceline Amoako [4]

1   Institut de Recherche Robert-Sauvé en Santé et en Sécurité du Travail (IRSST), 505 De Maisonneuve Blvd. West, Montréal, QC H3A 3C2, Canada; amelia.leon@irsst.qc.ca (A.L.); ai-thuy.huynh@irsst.qc.ca (A.-T.H.); jessica.dube@irsst.qc.ca (J.D.)

2   Department of Anthropology, Université de Montréal, Montréal, QC H3T 1N8, Canada

3   School of Management, Université du Québec à Montréal (UQAM), Montréal, QC H2L 2C4, Canada

4   School of Public Health Sciences, University of Waterloo, Waterloo, ON N2L 3G1, Canada; ellen.maceachen@uwaterloo.ca (E.M.); samantha.meyer@uwaterloo.ca (S.M.); smajowicz@uwaterloo.ca (S.M.); mkcrouch@uwaterloo.ca (M.K.C.); jamoako@uwaterloo.ca (J.A.)

5   School of Rehabilitation, Université de Montréal, Montréal, QC H3T 1N8, Canada; marie.laberge@umontreal.ca

\*   Correspondence: daniel.cote@irsst.qc.ca

**Abstract:** The urgency of managing the COVID-19 health crisis in workplaces led to tensions, work overload, and confusion about preventive measures. This study presents a secondary analysis of qualitative data on paradoxes and double binds (PDBs) experienced by precarious essential workers in Canada who interacted with the public and their supervisors. Based on 13 interviews from a larger qualitative dataset, we examine how workers navigated public health recommendations and organisational demands during the pandemic. Findings reveal multiple organisational and managerial PDBs—both COVID-19-related and pre-existing—that contributed to psychological distress and compromised well-being. We argue that PDBs represent a significant occupational health hazard for precarious workers. Addressing these structural contradictions through proactive management strategies could help mitigate workplace tensions, reduce stress, and enhance resilience in both crisis situations and regular organisational contexts. Our study contributes to occupational health and safety (OHS) by underscoring the risks posed by PDBs and advocating for strategies to support vulnerable workers in navigating conflicting demands.

**Keywords:** COVID-19; working environment; occupational health; paradoxical situations; double-bind theory; management; vulnerable workers; conflicting demands; qualitative study; Canada



## 1. Introduction

In the wake of the global coronavirus disease 2019 (COVID-19) pandemic in March 2020, drastic changes were required in workplaces and among the general population to prevent, limit, and contain the transmission of the virus and the disease. Containment was imposed in many states, schools were closed, and only so-called essential sectors were allowed to continue production activities, subject to extensive protective measures. Where possible, teleworking was made compulsory. Managing work during the pandemic was challenging, especially as employers had to reduce occupational risks, including COVID-19, which was classified as a workplace illness in many laws. This

scenario presented substantial challenges in terms of work relations and management (Butterick & Charlwood, 2021). The abrupt shift to remote work demanded rapid adaptation from both employers and employees. Traditional management practices had to be rethought, as maintaining productivity and employee morale in a virtual environment required new strategies and tools (Azizi et al., 2021). Communication, collaboration, and supervision had to be managed through digital platforms, which often led to increased stress and a sense of isolation among employees (Galanti et al., 2021; Marsh et al., 2022). Recent research has expanded on these early insights by highlighting the role of remote work leadership in mitigating stress and promoting employee well-being. Studies have shown that leaders who demonstrate transparency, empathy, and flexibility in virtual settings can buffer the negative effects of uncertainty and organisational change on workers (Galanti et al., 2021; Marsh et al., 2022). Likewise, effective crisis management strategies—including timely communication, access to resources, and workload redistribution—are essential to prevent burnout, especially in frontline or public-facing roles (Azizi et al., 2021; Barboza-Wilkes et al., 2024; Subramony et al., 2022). These findings underscore the importance of supportive leadership styles in times of crisis, a point echoed in our data where workers' psychological distress was often intensified by managerial opacity or unrealistic expectations.

Moreover, employers faced the daunting task of ensuring that their workplaces were safe for those who could not work remotely. Implementing health and safety measures, such as social distancing, regular sanitisation, and the provision of personal protective equipment, became paramount (Subramony et al., 2022). This not only involved significant logistical planning but also added financial strain on many businesses already struggling with the economic impact of the pandemic.

Additionally, the recognition of COVID-19 as an occupational disease brought about new legal and ethical responsibilities for employers. They had to navigate the complexities of workers' compensation claims and ensure that affected employees received appropriate support and care. Balancing these responsibilities while maintaining business operations required a delicate approach and highlighted the need for robust crisis management frameworks (Manjula & Sindhura, 2021).

The pandemic also amplified existing inequalities within the workforce (Matisāne et al., 2021). Low-paid essential workers faced higher exposure risks, increasing demands for fair wages, and better conditions (MacEachen et al., 2022; Reid et al., 2021). Employers had to tackle these issues while maintaining an inclusive workplace amid uncertainty. Essential workers, often from lower-paid sectors, faced higher risks of exposure, leading to a greater emphasis on the need for fair wages and improved working conditions. Employers had to address these disparities while fostering an inclusive and supportive work environment during a period of unprecedented uncertainty. In such contexts, the emergence of paradoxical and double-bind situations became apparent, as revealed by an inductive qualitative study of low-paid frontline workers in Québec and Ontario (Canada). These workers often found themselves caught between conflicting demands and expectations. These apparent paradoxes and double-bind situations (PDBs) highlighted the complex dynamics at play and the urgent need for more equitable and supportive workplace policies and practices. This article offers a deeper analysis of the occupational situations reported during the early phases of the COVID-19 pandemic and their potential impact on satisfaction and quality of life at work by using the paradox and double bind (PDB) theoretical framework for secondary analyses. The advantage of applying a PDB lens to the original exploratory research is that it magnifies phenomena, which became apparent to the authors during re-reading of interviews and exchanges between members of the research team.

## 1.1. Management of Organisations in the Context of COVID-19

Working in a complex organisation requires workers to negotiate different logics and operating models and both workers and employers to foster a balanced work dynamic. It is also necessary for employees to develop autonomy and obtain recognition and support through management-facilitated procedures for exchange and communication (De Gaulejac, 2009; Dejours, 2000; Morin, 2008). However, workplaces may be riddled with conflicting demands (Duterme, 2008). For example, a task may be defined in terms of production quotas, but at the same time be bound by the values of autonomy, decision-making latitude, and rigour, in short, by the imperative of completing quality work, always in compliance with the health and safety standards in force (Kristman et al., 2016; Paškvan & Kubicek, 2017). The urgency of managing the COVID-19 health crisis in the workplace may have generated PDBs (or exacerbated a pre-existing or borderline situation), which are known to lead to confusion, stress, anxiety, and the frustration of not being able to meet both demands (Røhnebæk & Breit, 2021).

This work context may influence the decision to take sick leave or not, or to report COVID-19 symptoms or not, especially when the job is precarious and the worker does not have sufficient income or may face reprisals or reprimands if their absence slows or interrupts a service or induces work overload in colleagues who remain on the job. A precarious job refers to employment that is typically low-paid—often around or just above the minimum wage established in each province—offers few or no social benefits (such as health insurance or paid leave), and is usually short-term, temporary, or lacks job security. Workers in precarious jobs may also face additional vulnerabilities related to their migratory or residential status, housing instability, or language and cultural barriers, which can further restrict their access to rights, protections, and workplace support (Kreshpaj et al., 2020). Conflicting demands can blur the line between personal and professional responsibilities, making it challenging for employees to prioritise their health over work obligations (including protecting the health of clients). The balance and cohesion of the work dynamic can be quickly undermined or disturbed by sickness absence. The original aim of this study was to explore how precarious essential workers interacting with the public and their supervisors understood the situation, made choices, and navigated public health recommendations to mitigate COVID-19 contagion at work (Hopwood et al., 2024). Analysis of the data from this study brought to light additional issues that could be likened to conflicting demands, similar to what is known in the scientific literature as paradoxes and double binds, each of which involve situations where there is no appropriate response and where any action taken will lead to adverse consequences (Berti & Cunha, 2023). This is somewhat comparable to what is commonly called a catch-22 situation, used to describe a no-win scenario where you are "damned if you do" and "damned if you don't" (Røhnebæk & Breit, 2021), and highlighting the frustrating and often absurd nature of such situations.

## 1.2. Double Binds and Paradoxical Systems Within Organisations

The paradox and double-bind concepts emerged in the wake of communication theory, which was developed in the late 1950s to describe certain types of interpersonal interactions (Bateson, 1972). For decades, it inspired researchers and therapists in many social sciences and humanities disciplines who were concerned about potentially harmful or destructive communication patterns in families or within organisations (Gibney, 2006; Poole & Ven, 1989; Schuham, 1967; Tracy, 2004; Watzlawick, 1963). While not necessarily a cause of mental illness, Gregory Bateson argued that contradictory communication—where explicit, implicit, or arising through meta-message conflict—can cause distress, confusion, and mal-adaptive responses (Bateson, 1972). Long anchored in family therapy, double-bind theory was revisited in the 1980s and extended to describe the process by which organisations

were transformed into paradoxical systems (De Gaulejac, 2010; Hennestad, 1990; Poole & Ven, 1989; Tracy, 2004). The terms "paradox" and "double bind" are often mistakenly used interchangeably. Although closely related, they are distinct: paradox refers to the conflicting demands themselves, whereas double bind refers to the organisational context that prevents communication about the paradox and offers no way out for the person caught in it, hence its "toxic" and potentially disabling and pathogenic effect, especially when the situation is recurrent and forms a communication pattern (De Gaulejac & Hanique, 2015). In the work context, conflicting demands seem to be mostly related to the work pace: the intensification–quality paradox, acceleration–deceleration paradox, and control–autonomy paradox (Evenstad, 2018). These situations can be exacerbated or generated by oppressive power relationships when authority is exercised, for example, in a coercive, manipulative, or dominative manner, or when circumstances (the "system") favour the subjectivation of paradoxical norms and practices so that incongruences are rarely questioned (Berti & Simpson, 2021). They are rarely questioned because, considered individually, intensification and quality seem perfectly logical, meaningful, natural, and self-evident (e.g., in a system where everyone is encouraged to be productive and improve their performance) (De Gaulejac & Hanique, 2015).

Evenstad (2018) classifies workplace paradoxes into two broad categories: organisational paradoxes and managerial paradoxes. Organisational paradoxes refer to the structural and cultural elements that characterise an organisation, while managerial paradoxes refer to managers' specific actions and ways of doing things, which have a direct impact on workers and work teams. It can be defined as a situation in which the individual attempts to manage conflicting demands that persist over time, and which appear logical in isolation but are irrational or even absurd when combined (Smith & Lewis, 2011). The organisational dimension includes the structure of the company, the hierarchical organisation, its mission, its values, all its policies and procedures, and the technological tools made available to workers. The managerial dimension includes the style of leadership being practised, interactions with the workers, ways of communication, decision-making processes, methods of conflict resolution, and management/evaluation of performance (Evenstad, 2018). Inspired by Evenstad's theoretical construct, Table 1 provides a descriptive summary of possible paradoxes in the workplace.

**Table 1.** Types of workplace paradoxes (Evenstad, 2018).

| Paradoxes | Conflicting Demands | Description |
|---|---|---|
| **Organisational** | | |
| **Change–stability** | Established well-defined procedures vs. innovative risk-taking skills | Organisations may want to foster performance by well-established procedures but also encourage creative and innovative initiatives. |
| **Exploration–exploitation** | Being flexible, creative vs. centralised decision-making | Organisations may encourage exploration based on the underlying logic of flexibility, decentralisation, openness to novelty, and creativity, while emphasising efficiency, productivity, and centralised decision-making. |

**Table 1.** *Cont.*

| Paradoxes | Conflicting Demands | Description |
|---|---|---|
| **Managerial** | | |
| **Acceleration–deceleration** | Being productive and fast working in a slow organisation that slows the pace of work | Demand for high productivity (producing in the shortest possible time) from workers in an organisation that has introduced highly bureaucratic operating and decision-making procedures that slow its employees' work may be a prelude for the autonomy–control paradox. |
| **Autonomy–control** | A double logic of worker autonomy and management control over work activities | The organisation wants to rely on professional autonomy, flexibility, individual responsibility, and personal discipline, while at the same time exercising control over the resources allocated, working hours, work pace, and definition of tasks. |
| **Intensification–quality** | Dual requirement of efficiency (to produce more) and quality | An organisation may introduce procedures for managing and evaluating performance that can have an impact on the workforce (e.g., downsizing) and the pace of work (e.g., intensification). Maintaining high quality standards may counteract an increase in the work pace. |

Organisational and managerial paradoxes have been hypothesised as being closely linked to the kind of power relations in place, to whether power is exercised directly or indirectly, and to its manifestation, i.e., episodic (coercion, manipulation) or systemic (enactment of a rule that may sometimes be implicit or taken for granted) (Julmi, 2022). In other words, the paradox can be introduced into the pragmatics of communication when a task is requested, often with ethical and deontological premises (e.g., to provide the necessary care and empathy to a patient, to do what is necessary to prevent severe acute respiratory syndrome coronavirus 2 (SARS-CoV-2) infection and transmission), and with a second requirement that defines the nature of the first, in this case, to be productive and to respond to an institutional logic (e.g., meeting patient quotas, avoiding service disruption and possible overwork for workers if a co-worker has to isolate themselves to prevent an outbreak in the workplace).

The analysis was carried out in the context of workplace infection prevention among low-income workers who had to deal with the public, while grappling with multiple forms of employment precariousness such as temporary or part-time work, agency work, or

limited social benefits (Kreshpaj et al., 2020). In light of these preliminary insights, we conducted a secondary analysis of qualitative data from a larger study on essential workers' experiences during the early COVID-19 response. This approach allowed us to explore how paradoxical double-bind situations manifested in practice, particularly among workers in precarious job conditions.

## 2. Materials and Methods

The data analysed in this article come from a qualitative exploratory study conducted in Ontario and Quebec, Canada, from August 2020 to March 2021. The study involved a purposive sample of 72 individuals, combined with elements of snowball sampling (or chain-referral sampling), which is used when participants assist researchers in identifying other potential participants (Pirès, 1997): 40 public-facing low-wage workers, 16 managers/supervisors, and 16 key informants. The inclusion criteria for workers were (1) over 18 years old; (2) working in an essential sector (i.e., essential to preserving life, health, and basic social functioning) during the early stages of the COVID-19 pandemic (the first three months of the pandemic when drastic public health measures were put in place); (3) low-wage earners; and (4) having physical proximity to clients in order to deliver the service. Inclusion criteria for managers and key informants were (1) over 18 years old and (2) holding a management or supervisory position during the early stages of COVID-19 in an essential sector that hires precarious workers, or working in an organisation dedicated to defending workers' rights or promoting occupational health and safety.

The guide for interviews with workers, managers, and key informants covered various topics related to COVID-19 mitigation measures in the workplaces and workers' exposure (see Table 1 for details). The interviews ranged from 60 to 120 min in length. They were conducted in English ($n = 36$), French ($n = 33$), or Spanish ($n = 3$) from August 2020 to March 2021. The open-ended interviews allowed sufficient time for participants to raise any other issues they considered relevant to the researchers' understanding, and which went beyond the strict framework of workplace health, such as work organisation, management, and labour relations. Interviews were recorded and transcribed by a professional transcriber using a word-processing software, and transferred to qualitative analysis support software NVivo12 for coding and to facilitate inference, interpretation, and theorisation (Clarke et al., 2015; Miles et al., 2018).

Data were analysed using insights from situational analysis (Clarke, 2003), a variation of grounded theory and an inductive set of methods aimed at developing theories by classifying and establishing logical associations between variables and emerging themes (Strauss & Corbin, 1998). This initial level of analysis highlighted issues relating to the enforcement of public health measures and how they placed these workers in stressful work situations due to additional disinfection tasks, protocols, and fear of possible adverse or even aggressive reactions from clients (Hopwood et al., 2024). We analysed 72 interviews. In 13 transcripts (7 workers, 2 managers, 4 key informants), we found examples of paradoxical or double-bind situations. These revealed an interrelationship between individual, organisational, and systemic factors that were likely to have influenced the implementation of public health measures aimed at reducing the risk of SARS-CoV-2 or COVID-19 infection and potentially revealing an adverse pre-pandemic workplace dynamic. No code existed for indexing the interviews based on the presence of a double bind. The identification of emerging themes such as paradoxical or double-bind situations constituted a more advanced phase of the analysis where connections were made by re-reading the data. The interview excerpts presented in this article fall under several index codes, including employment conditions, working relations, what is risky, plan vs. practice, reaction to risk measures, taking or not taking sick leave, and organisational changes needed. Following

thematic analysis, a pattern in some data was observed, revealing instances of workers facing conflicting demands. We conducted a second analytic search to deductively identify double-bind or paradoxical situations for workers. This approach ensured the exploration of concepts that are not common across all data but offer some meaningful insight into the experiences of a subset of workers. The subset was selected based on the presence of narrative elements or accounts that suggested tension, contradiction, or ambiguity in workers' experiences—elements that aligned with the concept of double binds. Rather than being predetermined by socio-demographic criteria, this subset emerged through an initial inductive coding process, during which we identified cases that appeared to reflect paradoxical expectations or conflicting demands. These cases were then revisited in the second, deductive phase of analysis to deepen our understanding of how such dynamics manifested in specific contexts.

The following section presents our key findings, beginning with the socio-demographic profile of participants, and then outlining the various paradoxical and double-bind situations identified in their narratives.

## 3. Results

This section presents the main findings, starting with the participants' socio-demographic characteristics. These are followed by a description of paradoxical or double-bind situations encountered in their workplaces and classified here as to whether they were induced by the COVID-19 context or not. The possible effects of double binds observed in the workplace and their impact on the emotional reactions of those who found themselves in these situations are reported throughout this section.

### 3.1. Socio-Demographic Characteristics of the Participants

A general profile of the participants (employees, managers, and key informants) is provided in Tables 2–4, respectively. Most workers and managers were employed in the retail sector (*n* = 22) and health and social services (*n* = 15), while most of the key informants were union and labour representatives (*n* = 5) or public health workers (*n* = 4). All but one of the workers were in direct contact with the public. Most workers were women (65%), and despite two unknown or missing pieces of information on origin, roughly equal numbers of workers were Canadian-born or immigrants (permanent residents). Most workers (25/40) had a college or university degree and about 18% had two or more jobs to supplement their income. Half of them reported being part of a "visible minority"[1].

**Table 2.** Guide for interviews with workers, managers, and key informants.

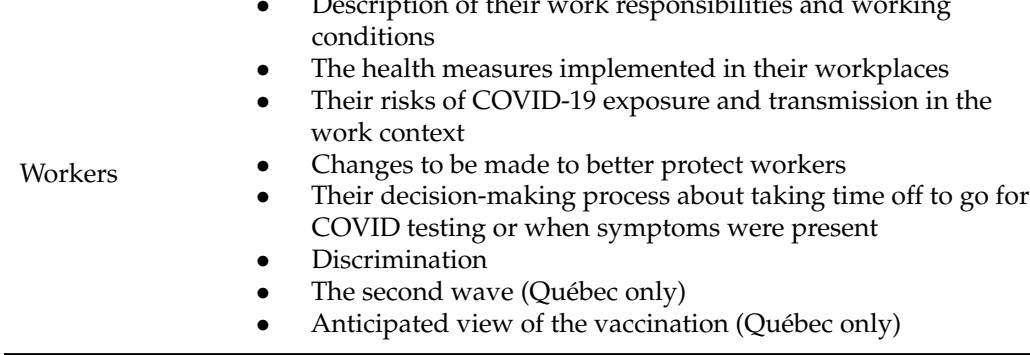

| | |
|---|---|
| Workers | • Description of their work responsibilities and working conditions<br>• The health measures implemented in their workplaces<br>• Their risks of COVID-19 exposure and transmission in the work context<br>• Changes to be made to better protect workers<br>• Their decision-making process about taking time off to go for COVID testing or when symptoms were present<br>• Discrimination<br>• The second wave (Québec only)<br>• Anticipated view of the vaccination (Québec only) |

**Table 2.** *Cont.*

| Managers or key informants | <ul><li>Main health risks for their employees or the populations they were assigned to protect</li><li>Management of COVID-19 in the workplace</li><li>Challenges of workers returning to work after a COVID-19 absence</li><li>Frequency of leave requests</li><li>Changes needed to better protect essential service workers</li><li>Issues faced by low-wage public-facing workers</li></ul> |
|---|---|

**Table 3.** Worker profiles (n = 40).

|  | *Original Data* | *Data Analysed for This Article* |
|---|---|---|
| **Age** |  |  |
| ■ *<20* | 2 | 0 |
| ■ *20–29* | 13 | 2 |
| ■ *30–39* | 10 | 0 |
| ■ *40–49* | 5 | 2 |
| ■ *50–59* | 6 | 2 |
| ■ *≥60* | 4 | 1 |
| **Place of birth** |  |  |
| ■ *Canadian-born* | 19 | 3 |
| ■ *Immigrant* | 18 | 4 |
| ■ *Missing information* | 2 | 0 |
| **Gender (self-reported)** |  |  |
| ■ *Male* | 14 | 1 |
| ■ *Female* | 26 | 6 |
| **"Visible" or racialised minority (self-reported)** | 20 | 2 |
| **Job type** |  |  |
| ■ *Cashier or customer service: supermarket, another food retailer, multi- or category retailer* | 16 | 1 |
| ■ *Social services* | 7 | 4 |
| ■ *Restaurant/café worker* | 4 | 0 |
| ■ *Customer service and/or sales: retail and repair shops, non-food* | 3 | 1 |
| ■ *Personal support worker* | 2 | 0 |

**Table 3.** *Cont.*

|  | | *Original Data* | *Data Analysed for This Article* |
|---|---|---|---|
| ■ | *Education services* | 2 | 0 |
| ■ | *Security guards and related security service occupations* | 2 | 0 |
| ■ | *Hairstylist* | 1 | 0 |
| ■ | *Machine operator and truck driver for a factory* | 1 | 0 |
| ■ | *School bus driver* | 1 | 1 |
| ■ | *Farmworker* | 1 | 0 |
| ■ | **University or post-secondary school degree** | 25 | 4 |
| ■ | **Having cumulative jobs (≥2)** | 7 | 0 |
| ■ | **Unionised** | 13 | 4 |

**Table 4.** Key informant profiles (n = 16).

| *Sector/Area* | *Original Data* | *Data Analysed for This Article* |
|---|---|---|
| *Unions and labour representatives* | 5 | 3 |
| *Public health* | 4 | 0 |
| *Workers' Compensation Boards* | 2 | 1 |
| *Legal clinics* | 2 | 0 |
| *Community organisation* | 2 | 1 |
| *Government* | 1 | 0 |
| *Sex or gender* | | |
| ■ *Male* | 8 | 1 |
| ■ *Female* | 8 | 4 |

*3.2. Paradoxical or Double-Bind Situations in the Workplaces*

Initially, the researchers identified 22 double-bind situations in workplaces, which were directly experienced or witnessed by 16 participants (8 workers, 6 key informants, 2 managers). These occurred mostly in the health and social services sector (N = 15 situations), surpassing those observed in retail trades (n = 3 situations), educational services (N = 3 situations), and passenger transport (N = 1 situation) (see Table 5). What created the double bind in the reported narratives was, firstly, the obligation to ensure one's own health and safety and protect the health and safety of others, and secondly, the obligation to perform, be efficient and productive, and avoid service disruption (e.g., the constant need to put on and take off countless pieces of personal protective equipment (PPE) such as gloves, masks, visors, and aprons). The double bind lay precisely in the fact of not being able to choose between the two injunctions or discuss the way they were implemented. The workers in this situation were in a subordinate position, and the effects of this double bind varied in intensity depending on the interaction, management, and power dynamic context. Table 6 identifies two classes of double-bind situations: those induced by COVID-19 and

those that were not. The situations that did not appear to have been induced by COVID-19 (N = 13) revealed a problematic work environment that pre-dated the pandemic (which the pandemic seems only to have amplified) and which was likely to persist even after the end of the health emergency. This represents a major workplace health issue in terms of psychological health at work (e.g., mental fatigue, anxiety, a sense of injustice, a possible loss of meaningfulness at work), which is not conducive to developing a respectful, friendly, and safe work environment. This point is examined in more detail below.

**Table 5.** Manager profiles (n = 16).

| *Sector/Area* | Original Data | Data Analysed for This Article |
|---|---|---|
| ■ Health and social services | 6 | 2 |
| ■ Retail stores | 6 | 0 |
| ■ Transport | 1 | 0 |
| ■ Clothing | 1 | 0 |
| ■ Commercial cleaning services | 1 | 0 |
| ■ Fitness and recreational sport centres | 1 | 0 |
| **Sex or gender** | | |
| ■ Male | 8 | 1 |
| ■ Female | 8 | 1 |

**Table 6.** Situations involving paradoxical or double-bind issues in work activities.

| **Double Bind Connectedness to COVID-19** | **Types of Workplace Situations** | **N** |
|---|---|---|
| ■ COVID-19-induced double bind | Involving the characteristics of the job | 6 |
| | Related to the employee's disability | 1 |
| | Related to an imprecise request that suggests doing the opposite | 1 |
| ■ Non-COVID-19-induced double bind | Related to the relationship between intensification and quality of work, acceleration and deceleration | 3 |
| | Passive acceptance of existing practices, instrumentalisation, and labour exploitation | 3 |
| | Involving toxic leadership and imbalance in power relations | 8 |

*3.3. COVID-19-Induced Double Binds*

Nine COVID-19-induced double-bind situations were identified in this study. They involved elements that related either to the characteristics of the job, the employee's disability situation, or the relationship between the intensification and quality of work, attributable to the health requirements imposed by the pandemic.

### 3.3.1. Situations Involving the Characteristics of the Job

For example, some situations involved conflicting demands, i.e., having to follow safety measures (such as physical distancing) and the impossibility of complying with them, given the characteristics of daycare work with young children. One participant described this as a "feeling of absurdity" because the work of a childcare provider necessarily involves constant contact with children: these youngsters—all under five years of age—need physical care (changing nappies, blowing noses, washing hands, etc.) and psychological and emotional comfort (cuddling, being consoled), which implies close physical proximity. This feeling of absurdity and incongruence of the demands with the nature of the work, combined with the fear of catching COVID-19, led the respondent to consider changing jobs until the Department of Health and Social Services lifted the obligation to wear a mask for this job category. The worker was not denying that wearing a mask could be protective but found it challenging in this working environment. It was at odds with what she perceived as feasible. The paradox stems from the fact that the worker had to follow public health instructions (which she seemed to believe in at the time of the interview) to limit the risk of infection and transmission of the virus, while at the same time providing an efficient childcare service, which she felt did not allow for wearing a mask.

Another participant, a union representative in the field of education, stressed the impossibility of maintaining physical distancing in this sector, where staff have no choice but to accept living with the risk of catching the disease. This union representative also spoke of the pressure to comply with preventive measures, which many workers in the sector described as "unbearable":

> *I've often heard this from local workers that you just have to keep in mind that you won't be able to apply the physical distancing measures, no, the pressure was unbearable. You can't do physical distancing in a school.* (Situation 6, Patrick, union representative in the field of education)

Other participants working in the health and social services, education, retail trade, and school transport sectors preferred to "follow their professional instincts" to give priority to "the children's needs". The perceived effects of these double-bind situations were often a "feeling of absurdity", "injustice", "nonsense", and "powerlessness", and also "confusion over the prioritisation of tasks". According to Patrick, silence seemed to be the best strategy for many workers to maintain a high quality of work despite the increased risk of infection or disease transmission.

### 3.3.2. Situations Related to the Employee's Disability

Only one situation referred to the characteristics of the job. That was the case of Catherine, who had to stand close to people to read their lips because of her hearing impairment. In this situation, Catherine felt excluded from her team and thought of leaving her job until the employer agreed to provide transparent masks for her interlocutors.

> *'I'm a hearing-impaired person, so it's really unnatural to distance myself, because to hear I have to get really close, and then the employer shouted at me, telling me that I wasn't respecting the rules (. . .) the manager asked all my colleagues to wear masks, so it was as if I didn't exist anymore, you know, I couldn't follow what was going on in a team meeting where everyone was present. So, I said, "I can't hear you, so I'm leaving".'* (Situation 2, Catherine, employee with hearing impairment)

### 3.3.3. Situations Involving an Imprecise Request That Suggested the Opposite

One situation involved an imprecise request that implicitly suggested the opposite (this issue could not be investigated further in this study).

*'We do have the option of refusing to serve customers, but we are not encouraged to use that option, basically. So, and I just don't feel that, like, if someone was like—oh, actually, unfortunately, I don't feel like, safe or comfortable serving a customer, "Please leave,"—I don't feel that that decision would be supported by management or anyone higher than management because we don't want people to feel unwelcome or like they're being policed or anything like that, [like]"follow the rules or you don't get service" [...] it very much feels like, well, you can do this, but if you do, you know, you're gonna get talked to about how you handled the situation or it's gonna be recommended that you don't do it again—that kinda thing.'* (Situation 22, Kelly, a retail worker)

Deciding whether to respond to the explicit request or to what it seemingly implied was experienced as stressful and a source of fear, given that the latter could lead to sanctions, reprimand, or blame, even though the explicit request seemed the wisest from a health standpoint. The paradoxical request may have been unintentional, given the need to react quickly to counter COVID-19, or it may have resulted from the manager's communication style. If the manager was trying to trap the employee or lead them to blame themself, this situation would be one of toxic leadership.

### 3.3.4. Situations Related to the Relationship Between Intensification and Quality of Work

Situations involving intensification and quality of work illustrate the issue of accumulating new and various tasks to contain the spread of the virus, leading to an accelerated work pace in order to complete everything, while at the same time ensuring quality work. For example, Gaston, a retail worker, mentioned that it was hard to perform multiple tasks, such as replacing/setting up sanitising bottles, while also attending to customers. The varying number of working hours was also an important factor in how work shifts were organised, as there were staggered shifts. There was a need for designated workers, but according to another participant, Amber, it seemed like people had to assume more responsibilities due to limited staff capacity during the pandemic. As for Gaston (cited below), the pre-pandemic situation was already difficult in terms of staff shortages, and the pandemic only exacerbated this situation, especially when all the workers caught COVID-19 before Christmas. He managed a homeless shelter alone for 14 days with no help from public health:

*'We've been over the edge for years... and now you've added another layer. I can't deal with your layer, you know. I'm sorry, I'd like to, but I can't. I can't do it alone with the resources I have.'* (Situation 5, Gaston, manager of a homeless shelter)

All these COVID-19-induced situations suggest how much pressure was put on workers to provide a high standard of service. This pressure sometimes meant going against public health directives, thus provoking strong emotional reactions. While such problems remained isolated and resulted from the emergency and health crisis, and may have disappeared as soon as the health constraints were lifted, they nevertheless raise important issues concerning the protection of workers' psychological equilibrium in times of crisis, tensions, or changes in the workplace. While the context of COVID-19 may have generated PDBs, attention must be paid to the work contexts leading to their reappearance after the pandemic. Some clues to the nature of these work contexts can be found in the next section.

### *3.4. Non-COVID-19-Induced Double Binds*

Some situations described as double binds were evidently induced by COVID-19, whereas others appeared to be related to organisational structure that superseded COVID-19. Some of these situations resemble those in the previous category, except that they reflect a difficult work context and organisation, marked at times by abusive exploitation, by the antithetical combination of intensification and quality of work, and by

management styles and power relationships that may seem "toxic". These can be described as "double-bind organisations", used as a generic category when it encompasses a whole range of situations involving management and organisational patterns. These working conditions and organisations do not fall within the commonly accepted definitions of precarious employment; rather, they highlight workplace dynamics and relational patterns potentially present in different types of employment, precarious or not.

### 3.4.1. Situations Related to Combining Intensification and Quality of Work

This kind of situation relates to the dual imperative of work intensification and delivery of quality work. In this case, workers pointed to the incompatibility of the two demands, as the continuation of this pace could pose a threat to their health and safety. The double bind resulted from the obligation to meet both demands with no possibility of challenging the obligation, with no way out and no opportunity to discuss and expose the burdensome nature of the situation openly, without embarrassment or fear. The following quote from Catherine describes an additional relational context that could, in her view, place the blame for a failure to meet targets solely on the worker.

> *In the organisational pyramid, we know that if something goes wrong, it's the employee's fault. Or if something goes wrong, it's because the worker has done something wrong. It's the manager's fault. So, we know the system is set up that way. We're not going to tell [facility management] that it doesn't make sense for you to give people [only] one bath a week. They're going to tell us [that] "it's the staff's fault", "why didn't you do that?", [...] but that's a different world from COVID. But COVID, at the same time, highlights [...] all these problems.* (Situation 3, Catherine, employee with hearing impairment)

In this situation, saying nothing and doing nothing became a strategy for action, but a maladaptive one since it generated a feeling of frustration and absurdity.

### 3.4.2. Situations Related to Passive Acceptance of Existing Practices, Instrumentalisation, and Labour Exploitation

Passive acceptance of a situation that is considered abusive means that workers can become accustomed to poor working conditions. As this union representative expressed it, workers often accept the strict minimum prescribed in the prevailing labour standards.

> *... they can't live without their wages for long, so the struggle to improve their conditions is not an easy one. Then there's a resilience that develops when you have to put up with bad conditions. You end up getting used to it and accepting your lot by saying, well, you don't deserve to have proper protective equipment, and you don't have to fight for it [laughs]. You just fight for a break or the right to go to the bathroom.* (Situation 9, Patrick, trade union representative)

This situation can go as far as "fear may force you to self-exploit", as related by Alejandro, a key informant. The need for integration and financial necessity can reinforce the idea that problems should be avoided, even if some rights are not respected by the employer. He addressed the issue of the migration process, suggesting that migrants or newcomers are in vulnerable positions and thus less inclined to speak out and claim their rights.

> *So when you come, if you're like an immigrant who needs money, who needs to work to pay the lawyer, to continue the migration process, etc., you don't want to have problems with the employer [...] you need to integrate [...] into Canadian society. So the trick is not to create problems. [...] And with the pressure to do a good job and all that, you forget yourself a lot, a lot, you put your own health on hold. [...] And it's as if fear also forces you to exploit yourself [...] So that... I felt that [...] even if I was a human*

*rights defender, I wouldn't be able to say to my boss, "Hey, give me a good glove because this one is broken", and maybe he would listen to me if I asked him, but my own fear of reprisals, of not being a good worker and of being fired is stronger, you know?* (Situation 12, Alejandro, key informant, social assistance sector)

For this manager, the pressure at work during the pandemic was caused by the use of mandatory overtime to make up for the workforce shortfall attributable to preventive withdrawal or infection with COVID-19. This work overload created "a vicious circle", a new threat that a growing number of workers would go on sick leave:

*So when we have compulsory overtime, well, sometimes that creates [situations where employees go on] salary insurance. It's a vicious circle. [. . .] It puts a lot of pressure on people to stay at work. That's a risk at the moment with COVID [. . .] It's going to create a risk of injury because people are going to want to do more things faster so they can get through the day with less staff.* (Situation 20, Constance, health manager)

3.4.3. Situations Related to Possibly Toxic Leadership and a Power Imbalance

Situations involving possible toxic leadership or problematic power relations may also be part of a broader organisational framework that is strained (but not necessarily paradoxical), except that they point more specifically to forms of interpersonal interaction as the possible source of the double bind. For example, Elena, a hygiene and sanitation worker, said the following in situations 15, 16, and 18 below, and Alejandro, a voluntary worker in the social assistance sector, had the following to say in situation 11 below:

*About conflicting demands*

*Your boss says, "If you're feeling bad, you can tell us, you can tell us. . ." but on the other side of the coin there's this nasty, negative attitude, like, "You've got to come to work. That's it. [mimics her boss's voice]: "Work, work, work, I don't wanna hear that you're sick." And I don't feel comfortable saying I'm sick. And I have the impression that this personal feeling is shared by several of my colleagues.* (Situation 15, Elena, health sector employee)

*About power imbalance*

*For me, it's also part of the precariousness that I don't feel comfortable asserting my rights [. . .] because if I do, I miss a pay-day.* (Situation 16, Elena)

*We noticed that when the CNESST* [Commission des norms, de l'équité, de la santé et de la sécurité du travail, or Quebec Workers' Compensation Board] *arrived* [at the workplace], *the employer was there, the supervisor was there, the workers were there, and they* [the CNESST inspectors] *started asking questions. The workers are going to say yes, yes, all the time because the employer is there* [laughs] *because there's a fear that if my boss is there, I'm going to talk badly about my boss, isn't there? So it's necessary for the CNESST to create spaces, dialogues with workers without the presence of the employer, to have a clearer vision of what's going on, isn't it?* (Situation 11, Alejandro, voluntary worker, social assistance sector)

*About toxic leadership*

*[. . .] And we're going to work under stress because the head of department is watching you all the time, under stress because of all these stressful situations at work due to the pandemic, because you have to concentrate to do your disinfection tasks properly.* (Situation 18, Elena, health sector employee)

These situations may be seen as persistent or as reflecting a long-term work context. The pandemic only amplified already existing difficulties, one of several risks of occupational health and safety (OHS) vulnerability. Once the pandemic ended, these problems

seemed likely to remain. Strategies for overcoming paradoxical or double-bind situations were scarce and described in rather vague terms. The most frequently mentioned were "withdrawing from the group" or from an "incongruous situation" when faced with the impossibility of doing one's job or complying with health measures. When this withdrawal strategy was unlikely or unfeasible without suffering reprisals or loss of salary, "hiding the symptoms" or "keeping silent" was seen as a calculated risk, a risky bet to avoid a financial hazard in lieu of preventing a health hazard, as if workers had to make a choice between the two. Arguably, workers with no benefits, atypical working hours (part-time, contract work, temporary work), and very little sick leave were more susceptible to this type of dilemma.

Despite limited direct insights from the field on this point, one case in particular—that of Elena—suggests that these dynamics may also reflect asymmetric power relationships and potential abuse grounded in racial or ethnic discrimination, intersecting with her precarious employment situation.

These findings highlight how paradoxical expectations and power imbalances were experienced at both structural and interpersonal levels. We now turn to a broader theoretical discussion to interpret these patterns and their implications for occupational health and organisational dynamics.

## 4. Discussion

As mentioned earlier, the aim of this article is to illustrate our hypothesis on the double-bind theory and the emergence of this form of paradoxical communication in workplaces during the early phases of COVID-19 in Canada (specifically, Ontario and Quebec). Analysis of the various health-related instructions disseminated and transmitted and the continuous and evolving changes in preventive practices within organisations during the early phases of the pandemic revealed several double-bind or managerial paradox situations. In some organisations, the pandemic context created different types of managerial paradoxes, while in others, it merely highlighted and exacerbated existing management and organisational challenges. This is the case in public institutions which, subject to relatively rigid and often centralised management mechanisms, pervert their original mission (such as health, education, and security) of adapting to client specificities (and to the complexity of the various life trajectories), while simultaneously achieving a certain standardisation of practices and performance measures (De Gaulejac & Hanique, 2015; Hartzband & Groopman, 2016; Hunt et al., 2019; Moraros et al., 2016). From the moment that the organisation produces and communicates values that contradict those of the individual, there is a sense of loss of meaning, demotivation, and dissatisfaction at work. This type of situation can lead to conflicting values and considerable psychological distress. In some contexts, where the gap between the practical demands of the job and the tasks prescribed by the organisation is very wide, this can lead to compassion fatigue. This seems to be the case particularly when the nature of the work involves alleviating the suffering of others and working with a vulnerable clientele (Côté & Dubé, 2019; Côté et al., 2021). Rather than seeing this as the sole effect of dehumanisation, which is tantamount to placing blame on the worker individually, it can be seen as a defence mechanism, where the worker does not have the latitude to use the strategy that seems most appropriate and best adapted to the needs of their clientele, as has been observed in the health and social services sector.

This internal conflict could then be mitigated by a highly developed sense of work or work ethic (Albert-Cromarias & Dos Santos, 2020; Morin, 2008) linked, for example, to the organisation's mission, if a "sense of work" is lacking. Sense of work refers to 'the meaning ascribed to it' and may encompass a wide range of elements, including job satisfaction,

meaning and purpose, engagement, or work ethic (Morin, 2008). A higher sense of work, as a protective buffer, would enable workers to intensify their performance, the production imperative being a command from within (Dardot & Laval, 2010).

*4.1. Double-Bind Situations Involving Managerial and Organisational Patterns That May Generate Paradoxes*

This section synthesises our findings with existing theoretical frameworks on organisational paradoxes, psychological distress, and worker agency. The situations described by the workers in this study suggest the presence of different types of paradoxes. It is no longer sufficient to simply state whether a management paradox exists or not. The most recent data from the literature show the importance of identifying the types of paradoxes that may be present in an organisation and describing their different modes of operation and potential effects, at both the personal level (psychological effects, emotional response) and the organisational level (work dynamics, team building, absenteeism, etc.). The typology outlined by Evenstad is useful for analysing the types of paradoxes and furthering the theoretical construction process (Evenstad, 2018). Identifying paradoxical situations is important because it could prevent them from turning into a double bind. While the paradox may seem inevitable or permanent insofar as a point of equilibrium or self-regulation is possible and the managerial context allows it, in the double bind, there is no possible reconciliation as the worker is in a lose–lose situation. The double bind constitutes a dysfunction, a disempowerment dynamic, and a failure of the adaptive process (Berti & Simpson, 2021; De Gaulejac & Hanique, 2015). The double-bind situations associated with the toxic leadership style (involving a power imbalance) appeared to generate more psychological distress, which perniciously harmed workers' mental health by causing considerable suffering.

The analysis of pre-existing tensions prior to the COVID-19 crisis resonates with anthropologist Rosemary Harris's (1987) findings on how organisational structures influence workers' capacity for genuine autonomy. Even in highly standardised work environments, Harris illustrates that local practices and power dynamics shape employees' experiences, often constraining their autonomy and reinforcing their vulnerability to workplace paradoxes. Similarly, research on paradox management (Smith & Lewis, 2011) and organisational power dynamics (Berti & Simpson, 2021) highlights how conflicting managerial logics can create persistent structural contradictions that leave workers with no viable resolution. An anthropological perspective on workplace relations, as advanced by Harris, provides valuable insights into how organisations become "paradox-inducing systems", where workers are caught in conflicting imperatives that limit their agency. Resistance within the factories she studied did not always take the form of overt dissent but was often expressed through strategies of adaptation and circumvention of constraints—an observation echoed in studies on worker agency under managerial paradoxes (Evenstad, 2018; Julmi, 2022). Similarly, our findings reveal that workers confronted with double binds during the pandemic frequently developed informal strategies to manage stress and maintain their mental health. While these strategies were necessary for navigating a complex work environment, they also highlight a form of structural powerlessness, where individual coping mechanisms mask the absence of deeper organisational reforms. As Berti and Cunha (2023) argue, paradoxical workplace conditions can generate disempowerment by forcing workers to internalise contradictions that ultimately serve managerial control. These findings suggest, as Harris and others have observed, that workplace management cannot be fully understood through an economic lens alone but must also be situated within the broader context of power relations, paradoxical constraints, and social negotiation.

While this study did not explicitly measure mental health outcomes, the re-reading of interview transcripts revealed consistent expressions of emotional strain—such as frustration, helplessness, and anxiety—particularly in relation to paradoxical demands and the

absence of meaningful avenues for communication or resolution. These insights, though exploratory, suggest that organisational paradoxes and double binds may contribute to psychological discomfort and emotional exhaustion, especially in contexts where power dynamics prevent workers from voicing concerns or negotiating their roles. We also recognise the conceptual and methodological challenges in measuring paradoxical double binds. PDBs are not always easy to identify or articulate—they often manifest as a patterned dynamic within relationships rather than a one-time event or discrete stressor. Their effects may be cumulative, insidious, and embedded in organisational culture or communication norms, making them difficult to capture through standard mental health or occupational stress measures. This complexity reinforces the need for interpretive, qualitative approaches to identify and analyse these dynamics, while also pointing to the importance of developing tools better suited to capturing their psychological impact.

### 4.2. Finding a Way out

Building on the challenges discussed above, we explore potential organisational responses to paradoxical systems, drawing from communication theory and recent workplace health literature. Several paradoxical situations observed in this study appear related to the difficult and unpredictable COVID-19 context and to have been resolved within a relatively short period of time or as organisations adapted to the pandemic context. Nevertheless, the same public health emergency context has made it possible to identify paradoxes that are likely to lead to complications or suggest a difficult—and, in some cases, relatively unhealthy—pre-pandemic working context (and conceivably post-pandemic too). These situations deserve sustained attention in workplace health research, including studies in organisational management, human relations, humanities, and social sciences studies in health. The analysis of interpersonal interactions using paradox theory and its extension to the analysis of organisational and management systems provide a guide to the solutions to be adopted.

The theoretical model dates back to the early work of the Palo Alto School on the double bind in the 1950s, and remains relevant as it insists on the need to actually call the paradox a "paradox" and to open the way to dialogue, open communication, and a reflective approach to deconstructing the paradoxical mechanisms. It is important to understand why demands, whether explicit or implicit, are paradoxical in order to act effectively. The Palo Alto School called this process "metacommunication", i.e., communication about communication itself, about the way things are communicated in the organisation, and about the content of messages, requests, or performance expectations, for example.

It is also important to analyse how paradoxical systems are set up and how everyone contributes to their installation and reproduction. It would then be possible to unravel the threads, first by becoming aware of them and then by dismantling them in daily practice (De Gaulejac, 2010). A few prerequisites apply, namely, a flexible organisational framework and a benevolent management style that allows everyone the freedom to express the constraints perceived. This can create new spaces for expression, which can lead to the transformation of the organisation, what De Gaulejac called "creative resistance" (p. 233) or what Coutarel and his team called "operational leeway" and the "power to act" (Coutarel et al., 2015), pointing to the fact that workers do not simply endure the work environment: they resist, try, invent, and create spaces for regulating their activity to meet production targets without compromising their health.

While it may be difficult, if not impossible, to eliminate a paradox, given the legitimacy and relevance of the conflicting demands, adaptation is possible by seeking a point of equilibrium (Ancelin-Bourguignon, 2018). The point of equilibrium may be fragile and unstable, but it can be a tipping point towards an evolving double bind or

towards the self-regulation of the organisational system that defines the work process, possibly even integrating everyday tensions into a process of reflexivity and discussion (Detchessahar, 2013).

Identifying the nature of the paradoxical situation or double bind and of the power dynamics, as suggested by Evenstad (2018), Julmi (2022), and Berti and Simpson (2021), might help in finding the appropriate remedy. The corrective action and strategy may vary, depending on the level (e.g., interactional, managerial, organisational) where the issue needs to be addressed.

Available data do not provide sufficient evidence that precariousness directly leads to double-bind situations. Precarious jobs often involve exploitation and various abuses without necessarily being paradoxical, in a capitalist framework (Benach et al., 2019). However, systems with inherent contradictions might exacerbate issues in precarious employment due to intersecting vulnerabilities (Mousaid et al., 2016). Unlike traditionally employed workers, temporary or on-call workers—especially those unfamiliar with their rights and workplace language—lack the same ability to voice disagreement or concerns, which can be compounded by their limited familiarity with public institutions and regulations (Côté, 2024). Precariousness may not involve PDBs at all; yet, it may worsen the situation. It may drive a paradoxical situation towards toxicity, or towards DBs, and elevate creative tensions into harmful realms, as suggested by some authors (Alge et al., 2023; Coutarel et al., 2015; De Gaulejac & Hanique, 2015).

This article provides analyses and reflections on PDBs at work from an interdisciplinary perspective and a broad conceptualisation of workplace risks, which includes risks related to organisational and managerial factors. However, the scope of these analyses is limited because our study was performed via secondary analysis and thus issues were not deeply probed in all interviews. Nevertheless, that this pattern was evident in the naturally reported data suggests further investigation of conflicts as potential double-bind situations is warranted.

## 5. Conclusions: New Research Issues in OHS

Our findings reveal the structural nature of paradoxical demands in precarious work contexts, both during and beyond crisis periods. These insights point toward several areas for future occupational health research and practical interventions, as summarised below. The main objective of this study was to analyse, from the standpoint of different managers, workers, and OHS practitioners, the issues involved in implementing health measures to curb the spread of COVID-19. Considering the specific OHS issues affecting workers in precarious situations, the goal was to gain a better understanding of the underlying work dynamics that may hinder or, conversely, favour preventive measures (Hopwood et al., 2024). Secondary analyses revealed that several work contexts were conducive to generating paradoxical situations or double constraints, whether induced by COVID-19 or not. The question arose as to whether precarious employment situations are more likely to give rise to organisational/managerial paradoxes, regardless of both the types described in this article and the dimensions of precariousness involved in this paradoxical effect. While it is impossible to provide a complete answer at the moment, future research should examine if unequal and disproportionate abusive power dynamics are potentially at play—that advantage is taken of the vulnerability of low-wage workers with temporary status or little social protection, members of minority groups, or those with a poor knowledge of the working language, cultural codes, work norms, and regulations in force (Côté, 2024). It is plausible that the effect of mutism and inhibition may be exacerbated in these workers, as well as the reaction of fear, anxiety, and isolation when they see no way out yet are concerned about keeping their modest

livelihood. Avoiding these reactions by implementing a benevolent management style would make it possible to address workers' professional concerns and thus acquire methods for identifying and mitigating paradoxical situations or double binds. These findings underscore the need for future occupational health and safety research to move beyond strictly structural analyses and account for the social and cultural dimensions of power within organisations. Indeed, while the double binds identified in this study are partially exacerbated by precarious conditions, they are also embedded in historical management logics that warrant deeper exploration, particularly through ethnographic approaches inspired by Harris's methodology.

Based on our findings, we recommend that workplace policies include structured opportunities for open dialogue around conflicting demands, especially in settings with high employment precarity. Organisational health strategies should move beyond individual resilience and explicitly address structural sources of paradox, such as contradictory performance expectations or unclear communication. Businesses and organisational leaders can take concrete steps by fostering psychologically safe environments where concerns about conflicting demands can be raised without fear of reprisal, implementing joint problem-solving mechanisms with frontline staff, and regularly reviewing protocols for feasibility in practice. Management training should incorporate awareness of paradoxical dynamics and their psychological impact, with a focus on promoting "metacommunication" practices that allow workers to express concerns without fear of reprisal. Policies ensuring access to paid sick leave, protection from retaliation, and culturally and linguistically inclusive communication are also essential to mitigate the disproportionate burden of PDBs on vulnerable workers.

**Author Contributions:** Conceptualisation, methodology, and validation, D.C., E.M., S.M. (Shannon Majowick), S.M. (Samantha Meyer), J.D. and M.L.; formal analysis and investigation, D.C., E.M., A.L., A.-T.H., P.H., M.K.C. and J.A.; data curation, A.L. and P.H.; resources, D.C. and E.M.; writing—original draft preparation, D.C.; every co-author participated in writing—review and editing; supervision, D.C. and E.M.; project administration, D.C. and E.M.; funding acquisition, D.C. All authors have read and agreed to the published version of the manuscript.

**Funding:** This research was funded by Institut de recherche Robert-Sauvé en santé et en sécurité du travail, with grant number IRSST-2020-0060.

**Institutional Review Board Statement:** This study received ethical approval from the University of Waterloo Human Research Ethics Board (protocol certificate number: 42449). Informed participant consent was obtained verbally and recorded before a telephone or videoconference interview, in alignment with ethical guidelines. Given the public health measures in place during the COVID-19 pandemic, including restrictions on in-person contact, written consent was not feasible. The verbal consent process ensured that participants were fully informed and agreed to participate while maintaining compliance with the ethical principles outlined in the Declaration of Helsinki and the Tri-Council Policy Statement: Ethical Conduct for Research Involving Humans (TCPS) in Canada. All names mentioned are pseudonyms to preserve confidentiality.

**Informed Consent Statement:** Informed consent was obtained from all subjects involved in the study.

**Data Availability Statement:** The data supporting this study are not publicly available due to confidentiality agreements with participants and ethical restrictions approved by the research ethics committee. The dataset consists of personal testimonies from workers, employers, health professionals, and trade union representatives, collected under strict confidentiality and anonymity protocols. To protect the identities of participants, as well as the names of the companies, public agencies, and districts or regions involved, data cannot be shared.

**Acknowledgments:** The authors are grateful to the IRSST for its financial support (grant no. 2020-0060) and technical assistance in the completion of this research. Thanks go to Steve Durant (a

postdoctoral fellow at the University of Waterloo) and Antonela Ilic (research assistant at the School of Public Health Sciences, University of Waterloo), who contributed to this research. The authors' appreciation also goes to the participants who volunteered their time, knowledge, and experience for this study. Special thanks to the Confédération des syndicats nationaux (CSN) and the Immigrant Workers Centre (IWC-CTI) for their help in recruiting participants. Without their cooperation, this research would not have been feasible.

**Conflicts of Interest:** The authors declare no conflicts of interest.

## Note

1. Being part of a "visible minority" was self-reported information, equivalent to "racialised groups". Racialised groups or visible minorities are social constructs, and specific groups that are considered "racialised" or "visible" can vary by region and period. Some countries such as Canada have enshrined the term "visible minorities" in law but exclude members of First Nations or Aboriginal peoples (Employment Equity Act). These controversial categories refer to physical characteristics that may expose people to various forms of systemic discrimination, racism and disparities in areas such as education, employment, housing, justice and healthcare.

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
