# Peer review of "Tackling Paradoxes and Double Binds for a Healthier Workplace: Insights from the Early COVID-19 Responses in Quebec and Ontario"

_humans, doi:10.3390/humans5020012_

Round 1
Reviewer 1 Report
Comments and Suggestions for Authors
Please see attached document

Author Response
Response to reviewer 1
The authors address an important issue with potentially far-reaching research and policy ramifications. The manuscript is well-written and well-organized. For optimization, minor revisions are recommended, primarily in the Materials and
Methods section. It would be helpful to provide more details about the dataset used, including additional information on the parent study and whether ethical considerations were specifically addressed in relation to the secondary data including recruitment methods.
Answer: Thank you for this comment. We believe there may be a misunderstanding regarding the relationship between the date used for this paper and the “parent” study. While we used the same dataset, our analyses were complementary and focused on different issues such as managerial dimensions such as the concept of "PDB" —that were not addressed in the research device or in the interview grids. It is written on page 7 that “We conducted a second analytic search to deductively identify double-bind or paradoxical situations for workers. This approach ensured exploration of concepts that are not common across all data but offer some meaningful insight into the experiences of a sub-set of workers”.
A clear definition of precarious employment is needed, along with more descriptive information about those included in the secondary analysis.
Answer: we propose this additional explanation following the material and methods section, just before getting into the results (page 7):
The subset was selected based on the presence of narrative elements or accounts that suggested tension, contradiction, or ambiguity in workers’ experiences—elements that aligned with the concept of double binds. Rather than being predetermined by sociodemographic criteria, this subset emerged through an initial inductive coding process, during which we identified cases that appeared to reflect paradoxical expectations or conflicting demands. These cases were then revisited in the second, deductive phase of analysis to deepen our understanding of how such dynamics manifested in specific contexts.
Some conceptual clarifications would improve the understanding of the study’s target group. For instance, the criteria used to define or evaluate low wages should be further discussed and, if it’s feasible, analyzed in connection with precarious essential workers.
Answer: we provided more detailed information based on Kreshpaj (2020) already cited in the manuscript:
A precarious job refers to employment that is typically low-paid—often around or just above the minimum wage established in each province—offers few or no social benefits (such as health insurance or paid leave), and is usually short-term, temporary, or lacks job security. Workers in precarious jobs may also face additional vulnerabilities related to their migratory or residential status, housing instability, or language and cultural barriers, which can further restrict their access to rights, protections, and workplace support.
It would be beneficial to discuss the adequacy of the sample size for the secondary data analysis and if possible, to include more context about the participant selection in quotes.
Answer: Thank you for your suggestion. We would like to clarify that the identification of the PDB (paradoxical double-bind) situations emerged during an exploratory phase of the analysis. These insights were not anticipated in the initial coding but surfaced through an in-depth re-reading of the interview transcripts. As such, the secondary analysis did not aim for thematic saturation but rather sought to shed light on a complex and analytically rich pattern observed in a subset of cases. We think that the supplementary text we just added provide sufficient details.
Although the relationship between PDBs and mental health outcomes is mentioned, it is not explicitly measured. A more thorough explanation linking organizational paradoxes and psychological discomfort could enhance the study’s coherence.
Answer: Thank you for this insightful comment. We acknowledge that while our study did not explicitly measure mental health outcomes, participants’ narratives often alluded to psychological distress—such as feelings of helplessness, frustration, or burnout—in the context of paradoxical double-bind (PDB) situations. We will expand our discussion to more clearly articulate how organizational paradoxes can contribute to psychological discomfort, drawing on relevant literature to support the link between conflicting demands, perceived lack of agency, and mental strain. This addition will help strengthen the conceptual coherence between PDBs and their potential psychosocial impacts. Here is a suggested new paragraph to provide insights (near the end of discussion section 4.1, page 18) :
While this study did not explicitly measure mental health outcomes, the re-reading of interview transcripts revealed consistent expressions of emotional strain—such as frustration, helplessness, and anxiety—particularly in relation to paradoxical demands and the absence of meaningful avenues for communication or resolution. These insights, though exploratory, suggest that organizational paradoxes and double binds may contribute to psychological discomfort and emotional exhaustion, especially in contexts where power dynamics prevent workers from voicing concerns or negotiating their roles. We also recognize the conceptual and methodological challenges in measuring paradoxical double binds. PDBs are not always easy to identify or articulate—they often manifest as a patterned dynamic within relationships rather than a one-time event or discrete stressor. Their effects may be cumulative, insidious, and embedded in organizational culture or communication norms, making them difficult to capture through standard mental health or occupational stress measures. This complexity reinforces the need for interpretive, qualitative approaches to identify and analyze these dynamics, while also pointing to the importance of developing tools better suited to capture their psychological impact.
Despite the important conclusions drawn in the discussion section, a more specific articulation of practical recommendations for policy could be beneficial.
Answer: thanks for this comment. We think it brings a great opportunity to strengthen the conclusion. While your current conclusion provides thoughtful reflections and identifies future research needs, it would benefit from more explicit guidance for policymakers and workplace actors. We suggest that short paragraph at the end of the conclusion section 5:
Based on our findings, we recommend that workplace policies include structured opportunities for open dialogue around conflicting demands, especially in settings with high employment precarity. Organizational health strategies should move beyond individual resilience and explicitly address structural sources of paradox, such as contradictory performance expectations or unclear communication. Businesses and organizational leaders can take concrete steps by fostering psychologically safe environments where concerns about conflicting demands can be raised without fear of reprisal, implementing joint problem-solving mechanisms with frontline staff, and regularly reviewing protocols for feasibility in practice. Management training should incorporate awareness of paradoxical dynamics and their psychological impact, with a focus on promoting “metacommunication” practices that allow workers to express concerns without fear of reprisal. Policies ensuring access to paid sick leave, protection from retaliation, and culturally and linguistically inclusive communication are also essential to mitigate the disproportionate burden of PDBs on vulnerable workers.
Additionally, a more comprehensive analysis of how businesses could actively address paradoxical situations would strengthen the paper’s practical implications, even though the prospect of long-lasting organizational change is acknowledged.
Answer: Yes, this is right. The added paragraph in the conclusion section provides a more targeted policy implications to explicitly include how businesses or organizational leaders can actively address PDB issues.
Finally, please ensure that the abbreviation OHS is explained within the manuscript.
Answer: Yes, we made sure OHS appears with its full form the first time we use abbreviations.

Reviewer 2 Report
Comments and Suggestions for Authors
I think you can improve the theoretical contextualization by expanding on recent research related to remote work leadership, burnout prevention, and crisis management in occupational health to better situate findings within the literature.
The research design & justification can be improved. You can provide more clarity on the selection criteria for the 13 analyzed interviews and justify how these represent broader trends.
To improve the logical flow & organization, you can improve transitions between sections and ensure arguments flow cohesively from problem statement to findings and conclusions.
Manuscripts are often enhanced by visual representation. You might want to consider adding a conceptual model or table summarizing paradoxical workplace structures for improved clarity.
You should strengthen the discussion on how organizations and policymakers can mitigate paradoxes and double binds in workplace settings.
Author Response
Response to reviewer 2
Comments and Suggestions for Authors
I think you can improve the theoretical contextualization by expanding on recent research related to remote work leadership, burnout prevention, and crisis management in occupational health to better situate findings within the literature.
Answer: we think the best place to address this issue is in section 1 (introduction), particularly toward the end of the paragraph that discusses remote work and organizational responses during the COVID-19 pandemic. So, we added a small paragraph after the sentence ending with “…a sense of isolation among employees” (page 2), and the added references align well with the theme of remote leadership, crisis management, and burnout prevention:
Recent research has expanded on these early insights by highlighting the role of remote work leadership in mitigating stress and promoting employee well-being. Studies have shown that leaders who demonstrate transparency, empathy, and flexibility in virtual settings can buffer the negative effects of uncertainty and organizational change on workers (Marsh et al., 2022; Galanti et al., 2021). Likewise, effective crisis management strategies—including timely communication, access to resources, and workload redistribution—are essential to prevent burnout, especially in frontline or public-facing roles (Azizi et al., 2021; Barboza-Wilkes et al., 2024; Subramony et al., 2022). These findings underscore the importance of supportive leadership styles in times of crisis, a point echoed in our data where workers' psychological distress was often intensified by managerial opacity or unrealistic expectations.
The research design & justification can be improved. You can provide more clarity on the selection criteria for the 13 analyzed interviews and justify how these represent broader trends.
Answer: we propose this additional explanation following the material and methods section, just before getting into the results (page 7):
The subset was selected based on the presence of narrative elements or accounts that suggested tension, contradiction, or ambiguity in workers’ experiences—elements that aligned with the concept of double binds. Rather than being predetermined by sociodemographic criteria, this subset emerged through an initial inductive coding process, during which we identified cases that appeared to reflect paradoxical expectations or conflicting demands. These cases were then revisited in the second, deductive phase of analysis to deepen our understanding of how such dynamics manifested in specific contexts.
To improve the logical flow & organization, you can improve transitions between sections and ensure arguments flow cohesively from problem statement to findings and conclusions.
Answer: Thank you for this constructive suggestion. We revised the manuscript to improve the logical flow between key sections by clarifying transitions from the introduction to the methods, from the methods to the findings, and between the discussion and conclusion. We also added bridging sentences within the discussion to strengthen the cohesion between the theoretical analysis and practical implications. These changes aim to enhance the clarity and coherence of the manuscript’s argumentative structure.
See the details:
- End of the introduction and beginning of methods: In light of these preliminary insights, we conducted a secondary analysis of qualitative data from a larger study on essential workers’ experiences during the early COVID-19 response. This approach allowed us to explore how paradoxical double-bind situations manifested in practice, particularly among workers in precarious job conditions.
- End of Methods and the beginning of result section: The following section presents our key findings, beginning with the socio-demographic profile of participants, and then outlining the various paradoxical and double bind situations identified in their narratives.
- End of the results section and the beginning of the discussion: These findings highlight how paradoxical expectations and power imbalances were experienced at both structural and interpersonal levels. We now turn to a broader theoretical discussion to interpret these patterns and their implications for occupational health and organizational dynamics.
- Within the discussion, we considered inserting short topic sentences at the beginning of sub-sections to preview how they build upon each other. For exemple:
- At the start of sub-section 4.1: This section synthesizes our findings with existing theoretical frameworks on organizational paradoxes, psychological distress, and worker agency.
- At the start of sub-section 4.2: Building on the challenges discussed above, we explore potential organizational responses to paradoxical systems, drawing from communication theory and recent workplace health literature.
- End of the Discussion and the conclusion, in agreement with your general comment on logical flow, we think it could be slightly strengthened to emphasize the flow from findings to implications. So, we suggest this bridge at the beginning of section 5: Our findings reveal the structural nature of paradoxical demands in precarious work contexts, both during and beyond crisis periods. These insights point toward several areas for future occupational health research and practical interventions, as summarized below.
Manuscripts are often enhanced by visual representation. You might want to consider adding a conceptual model or table summarizing paradoxical workplace structures for improved clarity.
Answer: Thank you for this suggestion. While we agree that visual representations can often enhance clarity, we believe that in the case of this manuscript, the complexity and nuance of paradoxical double-bind (PDB) situations are better conveyed through detailed narrative analysis and contextualized examples. Given that PDBs are relational, dynamic, and embedded in workplace interactions, a static conceptual model or table risks oversimplifying the very tensions and ambiguities we aim to illuminate. That said, we have worked to ensure that Table 1 (Evenstad’s typology) and the structure of the Results section already serve as conceptual anchors to guide the reader through our analysis. We are open to revisiting this in future work where a model could be developed based on a broader dataset or additional theoretical elaboration.
You should strengthen the discussion on how organizations and policymakers can mitigate paradoxes and double binds in workplace settings.
Answer: thanks for this comment. We think it brings a great opportunity to strengthen the conclusion. While your current conclusion provides thoughtful reflections and identifies future research needs, it would benefit from more explicit guidance for policymakers and workplace actors. We suggest that short paragraph at the end of the conclusion section 5:
Based on our findings, we recommend that workplace policies include structured opportunities for open dialogue around conflicting demands, especially in settings with high employment precarity. Organizational health strategies should move beyond individual resilience and explicitly address structural sources of paradox, such as contradictory performance expectations or unclear communication. Businesses and organizational leaders can take concrete steps by fostering psychologically safe environments where concerns about conflicting demands can be raised without fear of reprisal, implementing joint problem-solving mechanisms with frontline staff, and regularly reviewing protocols for feasibility in practice. Management training should incorporate awareness of paradoxical dynamics and their psychological impact, with a focus on promoting “metacommunication” practices that allow workers to express concerns without fear of reprisal. Policies ensuring access to paid sick leave, protection from retaliation, and culturally and linguistically inclusive communication are also essential to mitigate the disproportionate burden of PDBs on vulnerable workers.

Round 2
Reviewer 1 Report
Comments and Suggestions for Authors
I would like to thank the authors for their revisions. I have no further comments.
Author Response
Thank you for your confirmation and for taking the time to review the final make-up. I truly appreciate your input and collaboration.
Reviewer 2 Report
Comments and Suggestions for Authors
The comments seem to have been addressed in a satisfactory way. My only other comment is that I would have liked to see a higher percentage of the references within 5-7 years, but the references provided are sufficient.
Author Response
Thank you for your feedback and for taking the time to review our revised version. We would add that the relationship between paradox and double bind (PDB) and health at work remains an emerging area, and not the primary focus of our expertise. Consequently, few studies address this connection directly. We agree that a more targeted investigation could benefit from a scoping review approach to better map the existing evidence and identify directions for future research. Alternatively, one could focus on developing an interview guide aimed at capturing paradoxical situations and double binds without naming them explicitly—possibly by exploring specific themes or variables such as autonomy, control, and others. Clearly, there is still much to explore in this area.